# Migration direction in a songbird explained by two loci

Kristaps Sokolovskis [1] ✉, Max Lundberg[1], Susanne Åkesson [1], Mikkel Willemoes[1], Tianhao Zhao[2], Violeta Caballero-Lopez[1] & Staffan Bensch [1]

Migratory routes and remote wintering quarters in birds are often species and even population specific. It has been known for decades that songbirds mainly migrate solitarily, and that the migration direction is genetically controlled. Yet, the underlying genetic mechanisms remain unknown. To investigate the genetic basis of migration direction, we track genotyped willow warblers *Phylloscopus trochilus* from a migratory divide in Sweden, where South-West migrating, and South-East migrating subspecies form a hybrid swarm. We find evidence that migration direction follows a dominant inheritance pattern with epistatic interaction between two loci explaining 74% of variation. Consequently, most hybrids migrate similarly to one of the parental subspecies, and therefore do not suffer from the cost of following an inferior, intermediate route. This has significant implications for understanding the selection processes that maintain narrow migratory divides.

Seasonal migration is a widespread strategy for animals to avoid temporally adverse conditions on breeding sites[1,2]. Individuals of many songbird species migrate solitarily to specific and remote locations, implying that instructions for precise orientation are inherited[3]. Previously, seminal work using selective breeding of migratory blackcaps *Sylvia atricapilla* concluded that migration direction is controlled by one or few strong effect codominant loci[4]. Since then, identifying the underlying genes has been a major goal in bird migration research[3]. Recent efforts combining tracking and genotyping of wild migratory birds have revealed putative associations between a handful of genetic markers and specific wintering sites and/or migratory orientation[5–8]. Because the identified genetic regions are specific to each study, it remains unknown to what extent migration direction in birds has a common genetic basis.

We have studied a migratory divide between two willow warbler *Phyllsocopus trochilus* subspecies that have drastically different migratory routes. The nominate subspecies *P. t. trochilus* breeds in western Europe and migrates toward SW to western Africa while the subspecies *P. t. acredula* breeds in northern and eastern Europe and migrates SE to eastern and southern Africa[9,10]. The breeding ranges of the subspecies overlap in central Scandinavia and in eastern Poland, where they form migratory divides consisting of hybrid swarms[11,12].

Genetically, *trochilus* and *acredula* are nearly identical with a genome wide $F_{st} = 0.001$[13]. However, inversion polymorphisms on chromosomes 1 (InvP-Ch1) and 5 (InvP-Ch5) containing ~140 and ~50 genes, respectively, and presence or absence of a large (≥12 Mb) repeat-rich block characterized by multiple copies of a specific transposable element motif (TE) not yet assigned to a specific chromosome (Migration Associated Repeat Block in *acredula* (MARB-a), have almost fixed differences between the subspecies[14,15]. Allele frequencies at these loci show coinciding geographic clines with the change in migration direction, as inferred from analyses of feather stable isotopes (Fig. 1a–d). Hence, we expect that InvP-Ch1, InvP-Ch5 and MARB-a contain or are placed near genes encoding the migratory strategies (e.g., direction and timing).

In total we deployed 466 geolocators on adult male willow warblers across Sweden (Fig. S1). We tagged 60 *acredula* and 50 *trochilus* in northern and southern Sweden respectively to establish migratory phenotypes of allopatric subspecies. The remaining 356 geolocators we deployed at the Scandinavian migratory divide. In total, we retrieved data from 72 individuals of which 51 are from the migratory divide. Each bird was blood sampled upon recapture and genotyped for InvP-Ch1, InvP-Ch5[16] and presence or absence of MARB-a[15].

[1]Department of Biology, Lund University, Ecology Building, SE-223 62 Lund, Sweden. [2]GELIFES, University of Groningen, Nijenborgh 7, 5172.0664, 9747 AG Groningen, The Netherlands. ✉e-mail: kristaps.sokolovskis@biol.lu.se

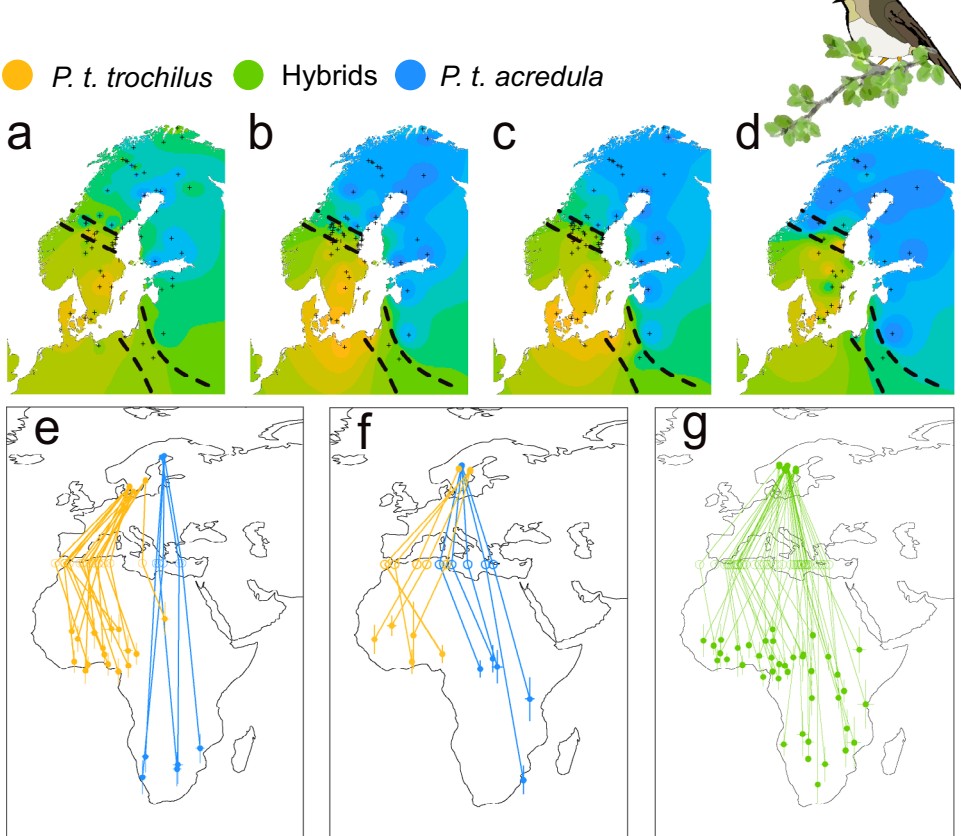

**Fig. 1 | Overview of the willow warbler system. a** Geographic structure of mean δ[15]N stable isotopes from winter-grown feathers ($n = 64$ sites). Color gradient shows extrapolation of δ[15]N values from low (orange) to intermediate (green), to high (blue). Low values associate with wintering in W-Africa, high with E-Africa. **b** Geographic structure for InvP-Ch1. **c** Geographic structure for InvP-Ch5. **d** Geographic structure for MARB-a based on a previous and less precise genotyping assay, AFLP-WW2. The data for **b** and **c** are from Lundberg et al.[13] and for **a** and **d** from Bensch et al.[12]. In **b**–**d**, progressing gradient orange-green-blue illustrates higher probability of birds carrying alleles fixed in *acredula*. **e** Tracks of allopatric *acredula* ($n = 5$) in blue and *trochilus* ($n = 16$) in orange. **f** Tracks of genetic *acredula* in blue (birds that have the MARB-a and are homozygotes for *acredula* alleles on invP-Ch1 and invP-Ch5, $n = 5$) and genetic *trochilus* in orange (birds that do not have the MARB-a and are homozygotes on invP-Ch1 and invP-Ch5 for *trochilus* alleles, $n = 5$). **g** Tracks of birds from the migratory divide with hybrid genotypes ($n = 41$). Hollow circles in **e**–**g** show estimated longitudes where birds crossed latitude 35°N (Mediterranean Sea). The lines connect individuals to their respective breeding site, Mediterranean crossing site and main winter site. Whiskers around locations in Africa in **e**–**g** show standard deviations in longitude and latitude from the main winter site. Crosses in **a**–**d** mark sampling sites. Source data are provided as a Source Data file.

Our results suggest that migration direction in willow warblers follows a dominant inheritance pattern. The *trochilus* allele at InvP-Ch1 is associated with a dominant effect for south-western migratory direction. The repeat-rich region (MARB-a) on the other hand is associated with a dominant effect for south-eastern migratory direction. We also find evidence for an epistatic effect of MARB-a suppressing the effect of InvP-Ch1.

## Results

During migration, both subspecies breeding in allopatry, i.e., *trochilus* ($n = 16$) from southern and *acredula* ($n = 5$) from northern Sweden, followed the expected routes to western and southern Africa, respectively (Fig. 1e). Migration routes of genetic *trochilus* and *acredula* breeding in sympatry in the migratory divide resembled those of their allopatric counterparts (Fig. 1f). The wintering locations of birds with mixed ancestry from the migratory divide were distributed across most of sub-Saharan Africa, with 23% of the sites located between the wintering quarters of the allopatric subspecies (Figs. 1g and S3). Analyses of tracks from the migratory divide (Tables S1–S3) in relation to the genotypes of the three genomic regions revealed that the presence or absence of MARB-a alone explains 62% of variation in autumn migration direction ($F_{49} = 82.4$, $p = 4.8 \times 10^{-12}$). The MARB-a effect is maintained

regardless of genotype combination on InvP-Ch1 and InvP-Ch5 (Figs. 2a and 3). Unlike the InvP-Ch1 and InvP-Ch5 that are chromosomal inversions the MARB-a is a repeat-rich block with a varying copy number of a specific TE[15]. Thus, our present genotyping protocol can only score for absence (<7 copies of TE) or presence (>7 copies of TE) of MARB-a (Fig. S2), the former assumed to be homozygous for the absence (−/−) and the latter either heterozygous (−/+) or homozygous (+/+) for the presence. To investigate whether the effect of MARB-a on migration direction was co-dominant or dominant, we use the TE-copy number as a proxy to separate between the two presence genotypes. Supported by the inheritance observed in two families (Table S5), those with inter-mediate copy numbers (7–19) are likely to be heterozygous (−/+) whereas those with many copies (>20) are likely to be homozygous (+/+). If MARB-a has a co-dominant effect, we expect a positive correlation between TE-copy number and migration direction, whereas no relationship is expected if the effect is dominant. There was no statistical effect of TE copy number on either winter long-itude ($R^2 = 0.0006$, $F_{1,22} = 0.01$, $p = 0.9$) or migration direction ($R^2 = -0.012$, $F_{1,22} = 0.7$, $p = 0.4$). This demonstrates that the effect on migration direction is caused by the mere presence of MARB-a rather than copy number per se, supporting the interpretation that MARB-a has a dominant effect on migration direction.

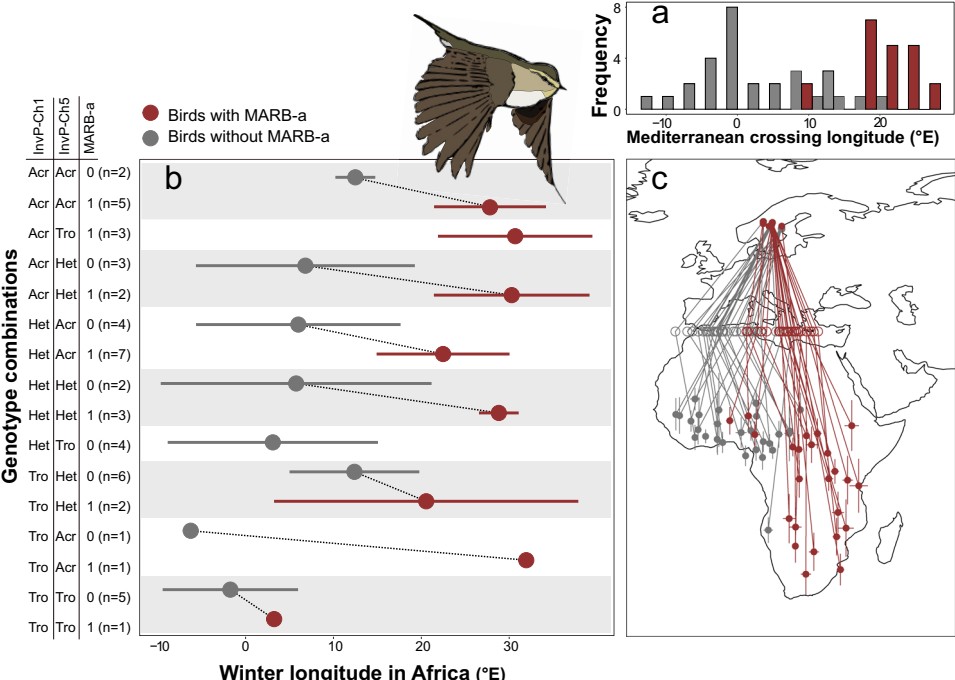

**Fig. 2 | The associations between African winter locations and genotypes of willow warblers tracked from the migratory divide. a** Histogram depicting bimodality in the frequencies of where birds from the migratory divide chose to cross Mediterranean. **b** Mean (whiskers depict ± standard deviations) winter longitudes for each of the nine combined genotypes of InvP-Ch1 and InvP-Ch5, illustrated separately for birds with (red) and without (dark gray) the MARB-a. On the *Y*-axis, the labels stand for genotype on InvP-Ch1, InvP-Ch5 (*Acr* homozygote for *acredula* allele, *Het* heterozygote, *Tro* homozygote for *trochilus* allele) and whether

MARB-a is absent "0" or present "1". Dotted lines connect the genotypes of both chromosomal inversions that have or do not have MARB-a. **c** Breeding and winter locations for birds from the hybrid zone with (red) and without (dark gray) the MARB-a. Hollow circles show estimates of longitudes where birds crossed the Mediterranean Sea. The lines connect locations of each individual. Error bars of locations in Africa show standard deviations in longitude and latitude of the main winter site of each bird. Source data are provided as a Source Data file.

The InvP-Ch1 had a noticeably smaller but still significant effect on migration direction ($F_{48} = 7.6$, $p = 1.4 \times 10^{-3}$, $R^2 = 0.2$, Figs. 2a and S4a). Interestingly, the *trochilus* allele on InvP-Ch1 appears to have a dominant effect that is epistatically suppressed by presence of MARB-a. Individuals that are heterozygotes or homozygotes for the *trochilus* allele on InvP-Ch1 tend to follow the western route unless they have the MARB-a (Figs. 2 and 3 and S5). A linear model including MARB-a, Invp-Ch1 and the interaction between these variables explains 74% of total variation in autumn migration direction ($F_{45} = 29.5$, $p = 3.6 \times 10^{-13}$). The functional roles of InvP-Ch1 and MARB-a in willow warblers remain unknown. Despite the geographic structure of InvP-Ch5 matching the locations of the migratory divides (Fig. 1d), it did not associate with migration direction in our data ($F_{48} = 1.8$, $R^2 = 0.03$, $p = 0.17$, Fig. S4b).

Autumn departure timing from the migratory divide did not associate with any of the three genomic markers ($F_{45} = 0.26$, adjusted $R^2 = -0.08$, $p = 0.93$). However, autumn departure date was positively associated with breeding latitude ($F_{71} = 16.29$, $R^2 = 0.08$, $p = 0.0001$), suggesting that in willow warblers, migration timing is either a polygenic additive trait that cannot be explained by the three markers and/or is to a large degree controlled by environmental factors. From the migratory divide, the great circle distance to the wintering range of allopatric *acredula* is about 50% longer than to the wintering range of allopatric *trochilus*. Because the birds wintering in West Africa cannot migrate more south than to the coast of the Gulf of Guinea (4°N), we restricted the analyses of genetic effects on migration distance to birds that wintered east of longitude 15°E. Almost all of these (21 out of 25) had the MARB-a and displayed large variation in migration distance (mean 7882 km: range 6122–10,200 km). However, neither invP-Ch1 nor invP-Ch5 had any effect on migration distance ($F_{16} = 0.41$, $p = 0.79$).

## Discussion

So far, the single most promising locus correlating with different winter longitudes in migratory passerines is the Vacuolar Protein Sorting 13 Homolog A (*VPS13A*) found in comparative genomic analyses of golden-winged *Vermivora chrysoptera* and blue-winged *V. cyanoptera* warblers[6]. Furthermore, a study on Swainson's thrushes *Catharus ustulatus* identified a ~30 Mb block on chromosome 4 containing a cluster of SNPs that was associated with migratory orientation in birds tracked with geolocators and in a separate group tested in captivity[5]. However, neither *VPS13A* nor any region on chromosome 4 showed as an outlier in genomic comparisons of the willow warbler subspecies[13]. Additional evidence that the avian chromosome 1 harbors genes of significance for migration has been recently reported from a study on common quails *Coturnix coturnix*, where an inversion polymorphism on chromosome 1 sets apart sedentary and migratory individuals[17].

Here we have found that in willow warblers, *P. trochilus*, a large proportion (74%) of the variation in migration direction was explained by a combination of MARB-a and InvP-Ch1, both with a dominant inheritance pattern and MARB-a suppressing InvP-Ch1 through epistasis. A consequence of this inheritance mode is that the migratory direction of most hybrids (76%, Figs. 1g and S3) resembled either allopatric *trochilus* or *acredula*. This contrasts with the pattern observed in European blackcaps, where F1 hybrids expressed intermediate orientations relative to the parental forms, suggesting that this trait is co-dominantly determined[4,8]. A codominant and additive inheritance pattern was also favored to explain the diverse migration routes of hybrid Swainson's thrushes[5].

Our study identifies two loci with strong effects on migration direction. The results presented here shed light on the genetic architecture of bird migration by showing that at least in some species

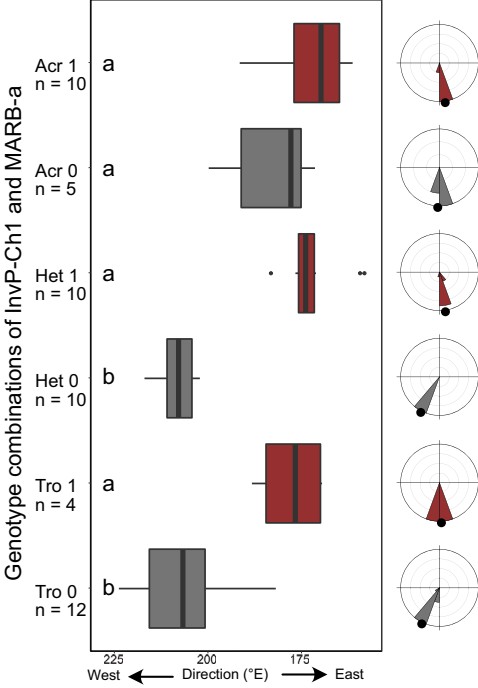

**Fig. 3 | Migration direction in relation to InvP-Ch1 and MARB-a.** Boxplots depicting bearing from breeding sites to the Mediterranean crossing (expressed in 0–360 degrees, centered to geographic north). The *Y*-axis shows the three genotypes of InvP-Ch1 (*Acr* homozygote for *acredula* allele, *Het* heterozygote, *Tro* homozygote for *trochilus* allele) in combination of whether MARB-a is absent "0" (gray) or present "1" (red). Letters to the left from boxplots show statistically different groups according to post hoc HSD Tukey test (Alfa = 0.5). In the boxplots, the boxes illustrate the lower and upper 25% quantiles with the median value as black bars. Whiskers show ranges. Outliers are considered as values more than 1.5 times the interquartile ranges and marked with black dots. Circular histograms to the right of the boxplots show distribution of migration directions for each corresponding group with black dots noting mean orientation. Source data are provided as a Source Data file.

migration direction follows a dominant inheritance pattern. Sutherland[18] suggested that genetic dominance might be one of the mechanisms that can explain the observed cases of rapid evolution of new migratory routes. A dominant inheritance pattern of migratory direction was also suggested by stable isotope analyses of winter grown feathers in hybrids of pied *Ficedula hypoleuca* and collared flycatchers *F. albicollis*, which indicated that hybrids were migrating similar to one of the parental species, the pied flycatcher[19]. Studies have shown that populations can evolve new migratory directions[20–22] within a timescale too short for de novo mutations to occur. These observations support our findings that alternative migratory phenotypes can remain hidden within recessive alleles and stay in populations at low frequencies until demographic events create recessive homozygotes. Moreover, that the willow warbler hybrids are more likely to follow one of the two parental migration routes instead of the possibly inferior intermediate route[4,23] has significant implications for understanding ecological and evolutionary processes that maintain narrow migratory divides. The classical assumption that migratory divides are maintained through higher mortality of F1 hybrids that take an inferior migratory route needs to be reevaluated.

## Methods

### Ethics statement

Animals' care was in accordance with institutional guidelines. Ethical permit was issued by Malmö-Lund djurförsöksetiska nämnd 5.8.18-00848/2018.

### Field work

We carried out the field work in Sweden during four breeding seasons (2018–2021). Adult male willow warblers were captured in their breeding territories using mist nets and playback of a song. From each bird, we collected the innermost primary feather from the right wing. From the birds that returned with a logger we also collected ~20 μl of blood from the brachial wing vein. The blood was stored in SET buffer (0.015 M NaCl, 0.05 M Tris, 0.001 M of EDTA, pH 8.0) at room temperature until deposited for permanent storage at −20 °C. We deployed Migrate Technology Ltd geolocators (Intigeo-W30Z11-DIP 12 × 5 × 4 mm, 0.32 g) and a nylon string to mount them on birds with the "leg-loop" harness method as outlined in our previous work[24]. The mass of the logger relative to that of the bird was on average 3.3% (range 2.7–3.8%).

The tagged birds were ringed with a numbered aluminum ring, and two, colored plastic rings for later identification in the field. In total, we tagged 466 males (349 in 2018 and 117 in 2020) at breeding territories. During the first tagging season (2018), birds were trapped at 17 locations (average 22 birds per site; range 7–30) distributed across Sweden (Fig. S1). Three of the sites were in southern Sweden to document migration routes of allopatric *trochilus* and three sites were located above the Arctic circle to record migratory routes of allopatric *acredula*, whereas the remaining (239) loggers were spread over 11 sites located in the migratory divide. Given the observed densities and distribution of hybrids after analyzing returning birds in 2019, we deployed 117 more loggers at one single site (63.439°N, 14.831°E) in 2020. We successfully retrieved tracks from 57 birds tagged in 2019 and 16 from birds tagged in 2021. In search for birds with loggers, we checked circa 3000 willow warbler males and covered an area of at least 0.5 km radius around each site the year after tagging.

### Geolocator data treatment

The R package *GeoLight* (version 2.0)[25] was used to extract and analyze locations from raw geolocator data. All twilight events were obtained with light threshold of 3 lux. The most extreme outliers were trimmed with "loessFilter" function and a *K* value of 3. We used GeoLight's function "getElevation" for estimating the sun elevation angle for the breeding period: these sets of locations were used to infer the positions for autumn departure direction. In addition, we carried out a "Hill-Ekström" calibration for the longest stationary winter site during the period before the spring equinox. Winter calibration produced location sets that better reflected the winter coordinates of the main winter site in sub-Saharan Africa[26]. We reduced some of the inherent geolocation "noise" by applying cantered 5-day rolling means to the coordinates. The equinox periods were visually identified by inspecting standard deviations in latitude. Latitudes from equinox periods were omitted (on average autumn equinox obscured data for 45 days (range 25–68). For the main winter site, we used the longest period at which bird stayed stationary and from which in all cases begun the spring migration (mean = 118, SD = 23 days). Timing of autumn departure was estimated by manual inspection of longitudes and latitudes plotted in time series. To estimate at which longitude the birds crossed the Mediterranean, we extracted the longitude when birds crossed latitude 35 N° (Mediterranean crossing longitude). For 29 birds, it was possible to directly extract the longitude at crossing latitude 35 N°. For the rest of the cases, the birds had not reached latitude 35 N° before the latitude was obscured by the equinox, we calculated the mean longitude of 10 days from the onset of fall equinox as a measure of the Mediterranean crossing. This measurement correlated highly with the winter longitude ($r = 0.78$, $p = 2.8 \times 10^{-16}$). To control for the birds relative breeding site longitude, we extracted the departure direction (1°–360°) relative from the tagging site to the location where the birds crossed the Mediterranean (departure direction). The departure data was of circular type (measured in 360°), however the variance did not span more than 180° degrees (range

151°–224°). Therefore, we proceeded with analyses using linear statistics. Geographic distances and departure direction were calculated using R package "geosphere" (version 1.5-10). Complete set of positions of each individual bird with equinoxes excluded is presented in Supplementary Data 1.

## Laboratory work and molecular data extraction

We extracted DNA from blood samples following the ammonium acetate protocol[16]. Genotyping for divergent regions on chromosome 1 (InvP-Ch1) and chromosome 5 (InvP-Ch5) was done using a qPCR SNP assay[16], which is based on one informative SNP per region (SNP 65 for chromosome 1 and SNP 285 for chromosome 5). Probes and primers were produced by Thermo Fisher Scientific and were designed using the online Custom TaqMan® Assay Design tool (Table S4). We used Bio-Rad CFX96™ Real-time PCR system (Bio-Rad Laboratories, CA, USA) and the universal Fast-two-steps protocol: 95 °C, 15 min–40*(95 °C, 10 s–60 °C, 30 s, plate read. Both regions contain inversion polymorphisms that restrict recombination between subspecies-specific haplotypes and contain nearly all the SNPs separating the two subspecies[13]. For each region, we scored genotypes as either "*Tro*" (homozygous for *trochilus* haplotypes), "*Acr*" (homozygous for *acredula* haplotypes) or "*Het*" (heterozygous). The method that we used to assess the presence of MARB-a is based on a qPCR assay that quantifies the copy number of a novel TE (previously known as AFLP-WW2[12]) that has expanded in *acredula*. The quantification of repeats by this method has been shown to be highly repeatable ($R^2 = 0.88$) when comparing estimates obtained from DNA in blood and feathers[15]. We used the forward (5′-CCTTGCATACTTCTATTTCTCCC-3′) and reverse (5′-CATAGGACAGACATTGTTGAGG-3′) primers developed by Caballero-López et al.[15] to amplify the TE motif. For reference of a single copy region we used the primers SFRS3F and SFRS3R[27]. We diluted DNA to 1 ng/µl⁻¹ and used a Bio-Rad CFX96™ Real-time PCR system (Bio-Rad Laboratories, CA, USA) with SYBR-green-based detection. Total reaction volume was 25 µl of which 4 µl of DNA, 12.5 µl of SuperMix, 0.1 µl ROX, 1 µl of primer (forward and reverse), and 6.4 µl of double distilled $H_2O$. We ran quantifications of the single copy gene and the TE variant found on MARB-a on separate plates with the following settings: 50 °C for 2 min as initial incubation, 95 °C for 2 min X 43 (94 °C for 30 s [55.3 °C SFRS3 and 55.5 °C for TE, 30 s] and 72 °C for 45 s). Each sample was run in duplicate and together with a two-fold serial standard dilution $(2.5–7.8 \times 10^{-2}$ ng). Allopatric *trochilus* have 0–6 copies whereas allopatric *acredula* have 8–45 copies[15]; a bimodal distribution was also confirmed in this new data set (Fig. S2). Accordingly, for the present analyses, we split the data in two groups: birds with ≤6 TE copies and birds with >7, translating into absence or presence of MARB-a, with the former assumed to be homozygous for the absence of MARB-a and the latter heterozygous or homozygous for the presence of MARB-a. Data from two investigated willow warbler families suggest a Mendelian inheritance pattern and provide support for our interpretation of how TE copy numbers reflect the three genotypes (Table S5). Moreover, the TE copy numbers within the hybrid swarm have a distribution similar to a combination of allopatric *trochilus* and *acredula*, further supporting that the copies are inherited as intact blocks (haplotypes). However, a precise distinction between heterozygotes and homozygotes on MARB-a is still not possible[15].

## Statistical analysis

We used linear models with departure direction, winter longitude, migration distance and departure timing as response variables and the three genetic markers: MARB-a (a factor with two levels), InvP-Ch1 (a factor with three levels) and InvP-Ch5 (a factor with three levels) as explanatory variables. Models were constructed with R base package "stats". We reported Type II ANOVA for models with more than one explanatory variable and no interactions and type III ANOVA results for models with interaction term by using R package "Car" (version

3.0-12)[28]. We initially constructed mixed effect models with timing of departure and tagging year as random factors however, this delivered singular fits due to insufficient sample sizes across categories. Normality of residuals was checked with a Shapiro–Wilk test. For carrying out circular statistics on autumn migration direction we used the R package "circular" (version 0.4-93). Watson's $U^2$ pairwise comparisons of different groups delivered the same results as linear models (Table S2 and Fig. S5). Circular means were identical to conventional linear means in our data set, which we take as another evidence that linear models are appropriate for the analysis of our data (Table S3 and Fig. S5). Maps in Figs. 1 and 2b and S1, S3 and S4 were created with R package "ggplot2" (version 3.3.6) using continent contours from Natural Earth, naturalearthdata.com/. Heat gradient over the maps in Fig. 1a–d were created with R package "gstat" (version 2.0-8) and the inverse distance weighting power of 3.0. Circular plots were created with ORIANA (version 4.02). All analyses were carried out with R version 4.1.1 (R Core Team 2021).

## Reporting summary

Further information on research design is available in the Nature Portfolio Reporting Summary linked to this article.

## Data availability

The geolocator data generated in this study have been deposited in the Dryad database under open access at https://doi.org/10.5061/dryad.stqjq2c6t. All other data underlying this study are provided in the Supplementary Information, Supplementary Data 1, and Source Data files. Source data are provided with this paper.

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

## Acknowledgements

We thank Harald Ris and Tore Dahlberg for invaluable help with the fieldwork, Charlie Cornwallis for advice on statistics and Sissel Sjöberg for comments on the manuscript. We are very grateful for funding from National Geographic Society WW-208R-17 (S.B.), Swedish Research Council 2017-03937 (S.B.), Royal Physiographic Society in Lund (K.S.), Royal Swedish Academy of Sciences (K.S.), Danish Council of Independent Research DFF-6108-00597 (M.W.) and Swedish Research Council 2016-05342 (S.Å.).

## Author contributions

Conceptualization: S.B., K.S., and S.Å. Tracking data analysis: K.S. and M.W. Laboratory work: V.C.L., T.Z., and K.S. Field data collection: K.S., M.L., S.Å., M.W., T.Z., V.C.L., and S.B. Writing—original draft: K.S. with input from S.B. Writing—review and editing: K.S., M.L., S.Å., M.W., T.Z., V.C.L., and S.B. Supervision: S.B.

## Funding

## Competing interests

The authors declare no competing interests.
