## [Peer Review File · Nature Communications]

REVIEWER COMMENTS

Reviewer #1 (Remarks to the Author):

The authors present an impressive and exciting migration study of willow warblers in Europe. This study system has become very important for the understanding of the evolution of migration, yet studies to date have not been able to follow the tracts of individual birds. The deployment of an impressive number of geolocator tags has resulted in a resolution of migration patterns in this species that has few comparisons. The combination of genotyping of genomic regions associated with the subspecies and migration patterns provides key insights into the strengths of linkages of these genomic regions and migration patterns. At first I was hoping to have seen genome sequencing from each of the individuals that has been individual tracked, however given the extensive previous genomic work it seemed like the present approach would be the most appropriate and it's difficult to imagine significantly more being gleaned from additional genome sequences from these birds.

I do have some minor comments that I think will improve the manuscript, but overall I think this is an exciting study with a fascinating result and an enjoyable read.

Line 12 - hybrid swarm or hybrid zone?

Line 14 - "migrated as one" -> "had a migration similar to"?

Line 16-18 - Unclear.

Line 38 - which chromosome is the MARB-a linked to? If not known, state clearly here. (On later reading I see it is not known the genomic location of this marker, but I think that needs to be stated clearly earlier).

Line 105 - "singling out" -> "identifying"

Lines 112 - 115 - I think this might be a bit too vague for the conclusion. For instance, if there are such "significant implications" it would be helpful to clearly articulate explicitly what those could be, or at least give some examples of how this information changes our understanding. I think this is an impressive finding, I just think this is a bit of a throwaway line that could use some more concrete language / examples.

Figure 1

- F: Make sure all the yellow and blue points are plotted over top of the green points. Also include a black outline for these individuals, as it's difficult to distinguish the green from the yellow specifically.

- Also, just an atheistic suggestion, but it might be nice to include an illustration of the bird(s) in the figures.

Figure 2 - I think the NN0 to SS1 notation to describe the genotypes useful, but it might also be easier to include a legend on the figure at the top or bottom indicating which letter corresponds to which marker.

Reviewer #2 (Remarks to the Author):

These authors combine tracking data from free-flying willow warblers with genotype data from a small number of genetic markers previously linked to migratory behavior in their species to provide additional detail behind the genetic mechanisms that control migratory behavior. Dominant inheritance patterns and epistatic interactions are found. This is an interesting study that uses a unique dataset. The resulting manuscript may be suitable as a short communication in Nature Communications but I have some hesitancy.

Specifically, we already know a lot about the genetics of migration in this system, including these specific genomic regions. How much are we really gaining here? The authors are still missing what I'd say is the holy grail of work on the genetics of behavior – the actual genes underlying variation. What might have changed my mind is if the authors had expanded beyond single genetic markers

to examine genome-wide variation. For example, they argue that they find no association between the genetic markers they have and timing because it's controlled by environmental factors. It seems highly unlikely to me that timing is controlled entirely by environmental variation. Perhaps if the authors had used genome-wide data they'd have identified a different genomic region linked to this migratory trait. That would have been an entirely new finding from the system. The genome-wide data could also help explain some of the missing variation. At first glance Figures 1 and 2 do not look like I'd expect dominant inheritance to be and that's likely because of the small changes afforded by genetic variants not examined in the present study. Given the fact these authors do a lot of work at the genome scale I imagine they are actually pursuing these angles right now.

Beyond the former hesitation, I have some concerns about how the authors have described (or not) previous research in the area. For example, some of the first work (outside European blackcaps) on migratory divides was done in flycatchers and found similar evidence for dominance (Veen et al. 2001). Why have the authors not mentioned this study? This seems like a pretty big oversight as it's a comparable European system. The authors also cite work conducted with Swainson's thrushes but are often incorrect in the details they provide. For example, Delmore et al. (2016 *Curr Biol*) did not only map wintering locations (line 27, 93), they used LLG just like the present study and mapped orientation. In addition, Delmore and Irwin (2014) did not find evidence for dominance (line 102). More accurate and complete details of previous work conducted in the field is needed.

Additional comments:

Why did the authors choose to limit their tracking data to where birds crossed the Med and the wintering location? Presumably they have a lot more location estimates than that? Perhaps it's related to accuracy but in that case please provide that information. Also how does changing the crossing point of the Med change their results?

Line 96 – I'd suggest the authors check that "chromosome 1" from the quail study is the same as the chromosome they are examining. Chromosome numbers got changed with the most recent B10K assemblies.

Figure 1 legend – should AFLP-WW2 actually be MARB?

RESPONSE TO REVIEWERS' COMMENTS

Reviewer #1 (Remarks to the Author):

The authors present an impressive and exciting migration study of willow warblers in Europe. This study system has become very important for the understanding of the evolution of migration, yet studies to date have not been able to follow the tracks of individual birds. The deployment of an impressive number of geolocator tags has resulted in a resolution of migration patterns in this species that has few comparisons. The combination of genotyping of genomic regions associated with the subspecies and migration patterns provides key insights into the strengths of linkages of these genomic regions and migration patterns. At first I was hoping to have seen genome sequencing from each of the individuals that has been individual tracked, however given the extensive previous genomic work it seemed like the present approach would be the most appropriate and it's difficult to imagine significantly more being gleaned from additional genome sequences from these birds.

Thank you for your encouraging words! We agree that additional genome resequencing data is not necessary.

I do have some minor comments that I think will improve the manuscript, but overall I think this is an exciting study with a fascinating result and an enjoyable read.

Line 12 - hybrid swarm or hybrid zone?

The term “hybrid zone” does not necessarily imply interbreeding beyond the F1 generation. By using the term “hybrid swarm” we wanted to convey that the hybridization also generates second generation hybrids and backcrosses.

Line 14 - “migrated as one” -> “had a migration similar to”?

Thanks, this is a better phrasing which we now have adopted.

Line 16-18 - Unclear.

We agree that the writing was not sufficiently clear. We have rephrased this part to make it clearer (lines 14-17).

Line 38 - which chromosome is the MARB-a linked to? If not known, state clearly here. (On later reading I see it is not known the genomic location of this marker, but I think that needs to be stated clearly earlier).

Thanks for pointing this out. We have now stated that the location of MARB-a is not known at the first mention of it in line 38.

Line 105 - “singling out” -> “identifying”

Done, now line 109.

Lines 112 - 115 - I think this might be a bit too vague for the conclusion. For instance, if there are such “significant implications” it would be helpful to clearly articulate explicitly what those could be, or at least give some examples of how this information changes our understanding. I think this is an impressive finding, I just think this is a bit of a throwaway line that could use some more concrete language / examples.

We have modified the end of the abstract (lines 14-17) and the discussion (lines 125 – 126) to support our statement.

Figure 1

- F: Make sure all the yellow and blue points are plotted over top of the green points. Also include a black outline for these individuals, as it's difficult to distinguish the green from the yellow specifically.

We have changed the fig. 1 and plotted the blue/yellow and green birds on separate maps.

- Also, just an atheistic suggestion, but it might be nice to include an illustration of the bird(s) in the figures.

Good idea! We have added willow warbler drawings to fig. 1 and 2.

Figure 2 - I think the NN0 to SS1 notation to describe the genotypes useful, but it might also be easier to include a legend on the figure at the top or bottom indicating which letter corresponds to which marker.

Done.

Reviewer #2 (Remarks to the Author):

These authors combine tracking data from free-flying willow warblers with genotype data from a small number of genetic markers previously linked to migratory behavior in their species to provide additional detail behind the genetic mechanisms that control migratory behavior. Dominant inheritance patterns and epistatic interactions are found. This is an interesting study that uses a unique dataset. The resulting manuscript may be suitable as a short communication in Nature Communications but I have some hesitancy.

We really appreciate your helpful insights and have carefully considered your suggestions to improve the flow and clarity of our manuscript.

Specifically, we already know a lot about the genetics of migration in this system, including these specific genomic regions. How much are we really gaining here?

In fact, until now there were only indirect associations between the three genetic markers (fig. 1b-d) and migration patterns inferred from stable isotopes (fig. 1a), i.e. birds from northern and southern Sweden had fixed differences whereas birds in the migratory divide consisted of

a diverse mix. This study takes an important step forward by investigating the association between migratory behavior and genotypes of individual birds. For the first time, we can quantify to what extent these three markers affect the migratory phenotype.

The authors are still missing what I'd say is the holy grail of work on the genetics of behavior – the actual genes underlying variation. What might have changed my mind is if the authors had expanded beyond single genetic markers to examine genome-wide variation.

We agree that the long-term goal in the field of migration genetics eventually is to pinpoint specific genes and mechanisms behind these fascinating behaviors. In this case, we argue that more resequencing data would not have provided much more knowledge about the focal questions raised in the manuscript (which of the previously identified three markers if any at all associates with migration direction?). We have already carried out extensive genomics work (Lundberg et al. 2017; Caballero-Lopez et al. 2022) to identify the parts of the genome that differ between the differentially migrating populations. As reviewer #1 noted, adding resequencing data is not likely to significantly improve the results.

For example, they argue that they find no association between the genetic markers they have and timing because it's controlled by environmental factors. It seems highly unlikely to me that timing is controlled entirely by environmental variation. Perhaps if the authors had used genome-wide data they'd have identified a different genomic region linked to this migratory trait. That would have been an entirely new finding from the system. The genome-wide data could also help explain some of the missing variation.

We agree that speculation on environmental factors is unnecessary here, so we have shortened this section and only kept the analyses that investigate migration-timing relative to the three studied markers (lines 74 -75). As pointed out by the reviewer #1, extensive resequencing from these birds and GWAS could be used to explore whether there are other regions in the genome associated to e.g., timing differences, but that is a different question and not the focus of the present manuscript.

At first glance Figures 1 and 2 do not look like I'd expect dominant inheritance to be and that's likely because of the small changes afforded by genetic variants not examined in the present study. Given the fact these authors do a lot of work at the genome scale I imagine they are actually pursuing these angles right now.

To help the reader to appreciate the bimodality of the Mediterranean crossing locations (i.e., a pattern in support of dominance), we have added an extra panel at the top of the fig. 2, a histogram that shows the strongly bimodal distribution of where birds from the migratory divide chose to cross Mediterranean Sea. The result of the analyzes that statistically demonstrates the epistatic interaction between the markers that are governed by dominant expression is illustrated in fig. 3. We hope that these changes will facilitate the reading of our manuscript.

Even though we did explain 74% of variation in migration direction, which for a study on a complex behavior in nature is very high, we agree that it is likely that other genomic regions are involved. With more comprehensive genomic data in the future, we might be able to explain an additional fraction of the remaining variation. However, such small effects probably require much larger sample sizes (several hundreds) than what we presently have.

Beyond the former hesitation, I have some concerns about how the authors have described (or not) previous research in the area. For example, some of the first work (outside European blackcaps) on migratory divides was done in flycatchers and found similar evidence for dominance (Veen et al. 2001). Why have the authors not mentioned this study? This seems like a pretty big oversight as it's a comparable European system.

We are aware of the paper by Veen et al (should be 2007 rather than 2001?) and that it sometimes is referred to as an example of dominant expression of migration direction. In the process of writing, we examined this paper in depth and concluded that the evidence for dominance is poorly supported by the data. Though the F1 hybrids are more similar to pied than to collared flycatchers in carbon stable isotopes, about a half of the hybrids had carbon isotope ratios within the zone of overlap between pied and collared flycatchers (see fig. 2 from Veen et al. 2007 below). We therefore opted to not cite the paper rather than criticizing someone's else's work directly.

We do however agree that it is an interesting observation and have now acknowledged that Veen et al. were the first to provide data suggesting dominant inheritance of migration direction (lines 114-118).

Figure 2. $\delta^{13}\text{C}$ and $\delta^{15}\text{N}$ isotope ratios for (a) pure pied (PF) and collared (CF) flycatchers and (b) hybrid individuals with maternal species in brackets.

The authors also cite work conducted with Swainson's thrushes but are often incorrect in the details they provide. For example, Delmore et al. (2016 Curr Biol) did not only map wintering locations (line 27, 93), they used LLG just like the present study and mapped orientation. In addition, Delmore and Irwin (2014) did not find evidence for dominance (line 102). More accurate and complete details of previous work conducted in the field is needed.

Thank you for pointing this out. We have now rephrased the text to give accurate references to the work by Delmore et al. (lines 27, 94-96, 108-109).

Additional comments:

Why did the authors choose to limit their tracking data to where birds crossed the Med and the wintering location? Presumably they have a lot more location estimates than that? Perhaps it's related to accuracy but in that case please provide that information. Also how does changing the crossing point of the Med change their results?

Yes, this is exactly the case, the geolocator data is too imprecise to be able to draw precise routes even outside the equinox periods. Delmore et al. 2016 did in fact summarize geolocator data from Swainson's thrushes as the longitude where birds crossed latitude 30°N. We took the same approach by describing where the birds chose to cross the largest ecological barrier *en route*. In our data the wintering longitude correlated strongly with the estimated Mediterranean crossing longitude ($r = 0.78$) giving extra confidence that the Mediterranean crossing point we have chosen is a good descriptor of the migratory route.

Line 96 – I'd suggest the authors check that "chromosome 1" from the quail study is the same as the chromosome they are examining. Chromosome numbers got changed with the most recent B10K assemblies.

Thanks for pointing this out. We have now double-checked that the divergent region assigned to chromosome 1 in the willow warbler matches chromosome 1 in the Japanese Quail.

Figure 1 legend – should AFLP-WW2 actually be MARB?

Up until now the MARB-a was only genotyped with an AFLP method that is less precise than qPCR genotyping. The heatmap provides extrapolation from previous work (Bensch et al. 2009) where the marker that we now know as MARB-a was known as AFLP-WW2. We have now included this explanation in the figure legend and the method section (line 198).

*All changes in the text are highlighted with yellow.